# Immunopeptidomic analysis of influenza A virus infected human tissues identifies internal proteins as a rich source of HLA ligands

Ben Nicholas[1,2]*, Alistair Bailey[1,2], Karl J. Staples[3], Tom Wilkinson[3,4], Tim Elliott[2], Paul Skipp[1]

1 Centre for Proteomic Research, Biological Sciences and Institute for Life Sciences, University of Southampton, Southampton, United Kingdom, 2 Centre for Cancer Immunology and Institute for Life Sciences, Faculty of Medicine, University of Southampton, Southampton, United Kingdom, 3 Clinical and Experimental Sciences, Sir Henry Wellcome Laboratories, Faculty of Medicine, University of Southampton, Southampton, United Kingdom, 4 NIHR Southampton BRC, UHS NHS FT, Southampton, United Kingdom

* bln1@soton.ac.uk

**Data Availability Statement:** The mass spectrometry proteomics data have been deposited to the ProteomeXchange Consortium via the

## Abstract

CD8+ and CD4+ T cells provide cell-mediated cross-protection against multiple influenza strains by recognising epitopes bound as peptides to human leukocyte antigen (HLA) class I and -II molecules respectively. Two challenges in identifying the immunodominant epitopes needed to generate a universal T cell influenza vaccine are: A lack of cell models susceptible to influenza infection which present population-prevalent HLA allotypes, and an absence of a reliable in-vitro method of identifying class II HLA peptides. Here we present a mass spectrometry-based proteomics strategy for identifying viral peptides derived from the A/H3N2/X31 and A/H3N2/Wisconsin/67/2005 strains of influenza. We compared the HLA-I and -II immunopeptidomes presented by ex-vivo influenza challenged human lung tissues. We then compared these with directly infected immortalised macrophage-like cell line (THP1) and primary dendritic cells fed apoptotic influenza-infected respiratory epithelial cells. In each of the three experimental conditions we identified novel influenza class I and II HLA peptides with motifs specific for the host allotype. Ex-vivo infected lung tissues yielded few class-II HLA peptides despite significant numbers of alveolar macrophages, including directly infected ones, present within the tissues. THP1 cells presented HLA-I viral peptides derived predominantly from internal proteins. Primary dendritic cells presented predominantly viral envelope-derived HLA class II peptides following phagocytosis of apoptotic infected cells. The most frequent viral source protein for HLA-I and -II was matrix 1 protein (M1). This work confirms that internal influenza proteins, particularly M1, are a rich source of CD4+ and CD8+ T cell epitopes. Moreover, we demonstrate the utility of two ex-vivo fully human infection models which enable direct HLA-I and -II immunopeptide identification without significant viral tropism limitations. Application of this epitope discovery strategy in a clinical setting will provide more certainty in rational vaccine design against influenza and other emergent viruses.

PRIDE partner repository with the dataset identifier PXD022884. All other relevant data is available in the manuscript and supporting information files.

**Funding:** This work received funding from Cancer Research UK Centres Network Accelerator Award Grant (A21998) to BN, AB, TE, PS and from Instrumentation in the Centre for Proteomic Research which is supported by the Biotechnology and Biological Sciences Research Council (BBSRC) (BM/M012387/1) to PS. https://www.cancerresearchuk.org/ https://www.ukri.org/councils/bbsrc/ The funders had no role in study design, data collection and analysis, decision to publish, or preparation of the manuscript.

**Competing interests:** The authors have declared that no competing interests exist.

## Author summary

Influenza infections present a significant global health challenge. High rates of mutation require reformulation of vaccines annually. Vaccines are designed to induce antibody responses to the surface proteins of the influenza virus, but the contribution of T cells to overall immunity is unclear. Here, we used several totally human laboratory models to show how the viral proteins are presented to the T cells to induce immunity. We found that CD8 T cells, which kill infected cells, and CD4 T cells which support the CD8 T cells as well as the antibody-producing B cells, mainly see proteins from inside the viral particle, not the surface ones which are targeted by antibodies. These internal viral proteins are more similar between different viral strains than the surface proteins, and therefore suggest that vaccines designed to induce T cell responses could be better protective if they target internal viral proteins.

## Introduction

Influenza virus is a major cause of morbidity, with every individual predicted to have 1–2 illness episodes per decade. There are approximately 1 billion annual cases of influenza globally, of which 3–5 million are severe, resulting in up to 650,000 deaths [1]. Additionally, the risk of a pandemic is ever-present, with likely further global costs of billions of dollars. There is widespread viral resistance to antiviral medications such as amantadine [2] and developing resistance against oseltamivir [3]. Both reduce symptom severity and duration but, critically, do not protect against primary infection and are least effective in at-risk individuals [4]. The most effective anti-influenza prophylaxis is vaccination which has, on average, 40–60% efficacy across all current strains [5].

Seasonal immunization efficiently generates neutralizing antibodies against viral haemagglutinin (HA) specific to the immunizing strain, however high mutation rates, particularly in the viral coat proteins haemagglutinin and neuraminidase (NA), rapidly cause antigenic drift leading to immune escape, requiring vaccines to be reformulated annually, based upon epidemiological predictions of the predominant global strains. Production and manufacturing of a clinically proven influenza vaccine is lengthy (>5 months) and costly.

Influenza vaccines designed to induce strong neutralizing antibody responses to haemagglutinin offer narrower and more short-lived immunity than naturally acquired infections, which also induce antibody responses predominantly to HA, but also stronger responses to NA and some internal viral proteins [6, 7]. Although neutralizing antibodies provide key protection against initial infection, T cells play an equally important role in limiting the consequent illness [8].

T cells recognise viral peptides bound to class I and II major histocompatibility (MHC) molecules which are presented at the cell surface. CD8+ T cells recognise endogenously processed viral peptides presented by class I MHC molecules on the surface of infected cells, whereas CD4+ T cells recognise exogenously processed peptides presented by class II MHC molecules mainly presented on the surface of professional antigen presenting cells such as dendritic cells and macrophages [9].

Targeting conserved viral protein sequences, which are more commonly derived from internal viral proteins, should confer greater vaccine induced cross-protection against multiple influenza strains, and early evidence in mice supports this [10]. Previous evidence has shown that the influenza virus nucleoprotein (NP) is a major target of immunodominant CTLs in

direct infections [11], and acid polymerase T cell epitopes are more abundant in mouse cross-presentation models, but matrix protein (M1) and the RNA-directed RNA polymerase catalytic subunit (PB1) also contain conserved immunogenic sequences [12]. Viral NP and M1 are also major targets for immunodominant CD4 T cell responses [13]. Human infection trials suggest that pre-existing influenza-specific T cells, particularly those recognising conserved epitopes of internal viral proteins, are central to limiting disease severity following experimental challenge with different influenza strains [14].

Infections stimulate both CD4+ and CD8+ T cell subsets, and optimal humoral and cellular immunity is dependent upon the activation of CD4+ T helper cells, which support CD8+ T cell function but can also themselves have effector functions [15,16]. Virus-specific CD4+ and CD8+ T cells specific for immunodominant influenza epitopes negatively correlate with disease severity and fever symptoms, respectively [17].

In animal models, long peptide vaccines designed to stimulate antibody and T cell responses have provided only minimal protection against infection, with limited evidence of symptom reduction [18]. Rationally designed T cell epitope targeted vaccines containing long peptide sequences from the extracellular domain of M2 (M2e) and NP have been tested in mice, but offered only limited to moderate protection with variable responses to each peptide [19]. This may arise because candidate T cell epitopes are commonly identified using machine learning based algorithms to predict binding of 9-mer or 15-mer peptides to specific HLA-I and HLA-II HLA allotypes, respectively. Whilst peptide affinity predictions are reasonably accurate, at least for HLA-I, there are multiple other factors that influence the true efficacy of T cell epitopes, including, but not restricted to, the abundance of the source protein available for presentation by infected cells, the biochemical nature and structural stability of the epitope, the suitability of surrounding residues to endosomal processing, and the secondary structure of the source protein. Due to differences in mouse and human MHC, humanised mouse models must be utilised to examine influenza T cell epitopes in humans, but are then restricted to the transgenic allotypes, usually HLA-A*02:01, the most prevalent global allotype.

Recent improvements in the sensitivity of mass spectrometry combined with immunoprecipitation of peptides bound to HLA-I and HLA-II have enabled the field of immunopeptidomics to be utilised in the search for optimal T cell epitopes [20]. Typically, influenza epitopes are identified from the elution of surface HLA-I associated peptides of an immortalised cell line infected with virus. [21]. Advantages of this approach are that the HLA-restricted peptide sequence of a known cell HLA allotype can be directly measured, showing that the peptide can be processed and presented, at least in-vitro. Such approaches have indicated protective immunopeptides across influenza A B and C strains [22].

A limitation of this immunopeptidomic strategy is the requirement for significant quantities of infected cell material (generally a minimum of $10^8$ cells). For this reason, previous studies have utilised cell lines which grow readily in the laboratory e.g. HELA cells, but which do not fully reflect in-vivo cell targets. This may not be of concern if cells are infectable, however where a virus shows strong infection tropism for a particular cell type, often the case in freshly isolated strains which have not been adapted to laboratory conditions, this could become a significant challenge. We have previously demonstrated that ex-vivo infection models of lung tissues are reflective of localised patterns of infection and subsequent inflammatory response, and can therefore be more accurately used to test respiratory viral inhibitors than cell models [23].

In the current study, we use this model to establish the antigenic landscape of human lung tissue 22 hours following influenza A virus (IAV) infection. By comparing immunopeptidomes recovered from infected explants with that of monocyte-derived dendritic cells (MoDC) following cross-presentation; and a directly infected monocyte/macrophages cell

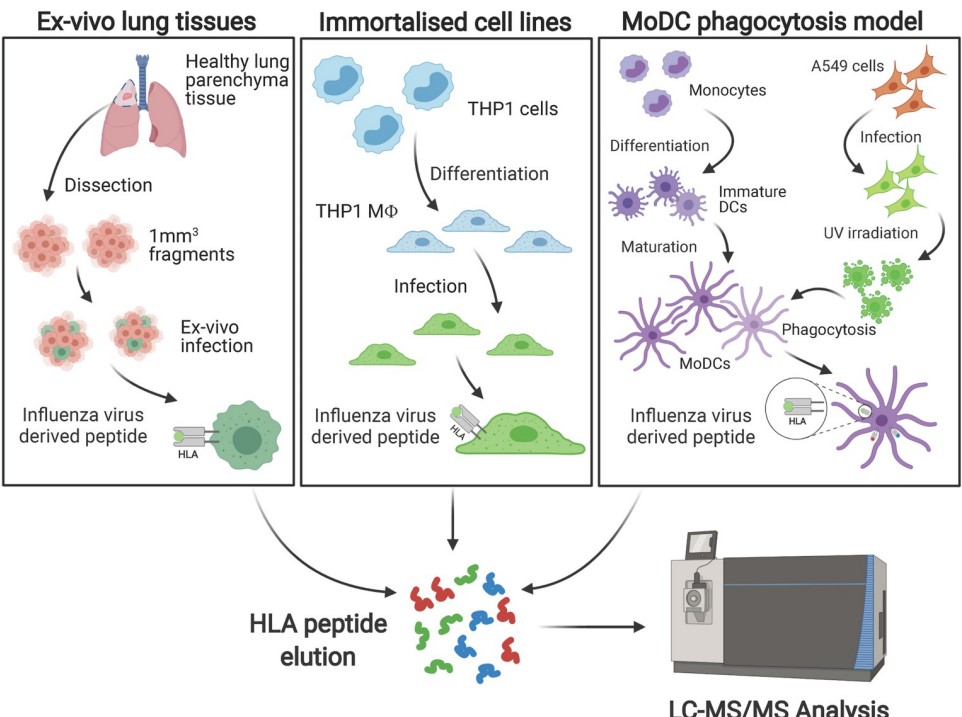

**Fig 1. Workflow of the approach to identify HLA-I and -II influenza immunopeptides isolated from cell lines, dendritic cells and lung tissues.**

line, we show that the local lung repertoire of IAV peptides available for immune recognition is dominated by HLA-I bound ligands sourced from internal viral proteins.

## Results

We used three different models as sources for immunopeptidome isolation, to identify HLA-I and HLA-II restricted influenza immunopeptides (Fig 1) [24].

### Infection of resected lung tissues reveals novel influenza HLA-I and–II restricted epitopes

We examined infection rates and HLA-presented peptides using ex-vivo lung tissue samples from three different human donors with diverse HLA types (S1 Table). Exposure of the ex-vivo lung parenchymal tissues: P1, P2 and P3 to the two viral strains studied herein led to variable infection rates in the two main cell types which have previously shown influenza susceptibility, epithelial and macrophage cells (Fig 2A) [23]. We have previously shown that epithelial infection rates in resected lung tissues can be variable [23], and this study indicates that the viral strain also affects infection rate in the two target cell types. Infection rates varied from 2%-70% in epithelial and 1%- 15% in macrophage cells.

Following ex-vivo Wisconsin H3N2 influenza infection, we observed 7944 distinct class I HLA peptides deriving from 2603 host proteins for P1, 5304 distinct class I HLA peptides deriving from 1696 host proteins for P2, and 6338 distinct class I HLA peptides deriving from 1996 host proteins for P3 (S2 Table). We also observed similar numbers for the same tissues infected with X31. We were able to identify a number of influenza-derived HLA-I restricted peptides across all samples derived principally from M1, NP and NS proteins (Table 1). Similar

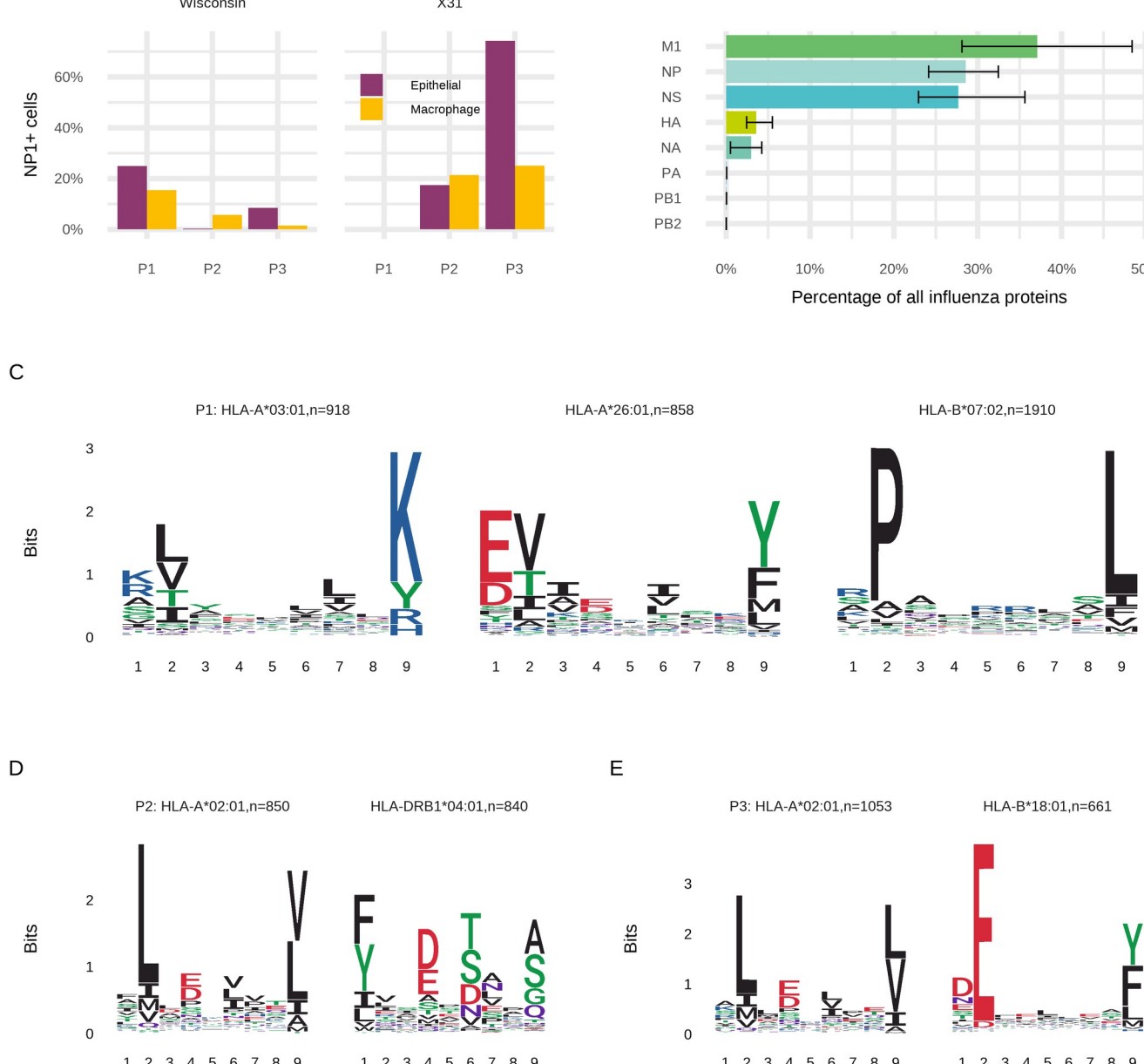

**Fig 2. Characteristics of the ex-vivo lung tissues used to identify viral HLA ligands. (A)** Infection susceptibility of epithelial and macrophage cells of ex-vivo lung parenchymal tissues for three samples to two influenza strains, A/H3N2/Wisconsin/67/2005 and A/H3N2/X31 **(B)** Relative proportions of influenza proteins present in the three lung tissue samples **(C-E)** Class I HLA allotype 9-mer binding motifs derived from host immunopeptides by unsupervised clustering using MixMHCp.

to previous findings, despite consistent, if limited, expression of viral haemagglutinin and neuraminidase in the proteome, HLA-I peptides were generally of internal viral protein origins: M1, NP and Non-structural protein 1 (NS1) (Fig 2B and Table 1).

Unbiased cluster analysis [25,26] was used to identify the binding motifs of all the distinct observed peptides for each sample (Fig 2C and 2E). These motifs indicated the respective HLA allotypes for each observed influenza peptide, which matched to the known HLA allotypes of

**Table 1. Summary of the HLA-I and II immunopeptides isolated from lung tissues infected ex-vivo with two influenza strains.** nM is NetMHC 4.0 predicted binding affinity.

| Protein | Protein Position | Length | Patient | Allotype | nM | ID | Wisconsin | X-31 |
|---|---|---|---|---|---|---|---|---|
| Matrix protein 1 | 243–252 | 10 | P1 | HLA-A*03:01 | 23.32 | IEDB_EPITOPE:54898 | RMGVQMQRFK | - |
| Non-structural protein 1 | 142–150 | 9 | P1 | HLA-A*26:01 | 18.52 | IEDB_EPITOPE:124082 | ETIVLLRAF | - |
| Non-structural protein 1 | 163–171 | 9 | P1 | HLA-B*07:02 | 21.2 | | LPSFPGHTI | - |
| Non-structural protein 1 | 122–131 | 10 | P1 | HLA-A*03:01 | 13.38 | | AIMEKNIMLK | - |
| Polymerase acidic protein | 516–525 | 10 | P1 | HLA-A*26:01 | 14.82 | | DVVNFVSMEF | - |
| Hemagglutinin | | 15 | P2 | HLA-DRB1*04:01 | - | | AADLKSTQAAINQIN | - |
| Matrix protein 2 | 7–15 | 9 | P2 | HLA-B*44:27 | - | IEDB_EPITOPE:68383 | - | VETPIRNEW |
| Non-structural protein 1 | 122–130 | 9 | P2 | HLA-A*02:01 | 23.35 | IEDB_EPITOPE:2022 | AIMEKNIML | - |
| Nucleoprotein | 305–313 | 9 | P2 | HLA-B*15:01 | 37.33 | IEDB_EPITOPE:54656 | - | RLLQNSQVY |
| Nucleoprotein | 404–412 | 9 | P2 | HLA-B*15:01 | 26.29 | | - | GQISIQPTF |
| Nucleoprotein | 450–458 | 9 | P2 | HLA-B*15:01 | 149.35 | | - | SARPEDVSF |
| RNA polymerase catalytic subunit | 177–185 | 9 | P2 | HLA-B*44:27 | - | | - | EEMGITTHF |
| RNA polymerase catalytic subunit | 94–103 | 10 | P2 | HLA-B*44:27 | - | IEDB_EPITOPE:16567 | - | FLEESHPGIF |
| Matrix protein 1 | 5–12 | 8 | P3 | HLA-B*18:01 | 107.2 | | TEVETYVL | TEVETYVL |
| Matrix protein 1 | 111–119 | 9 | P3 | HLA-B*15:01 | 244.02 | | GAKEIALSY | - |
| Matrix protein 1 | 111–119 | 9 | P3 | HLA-B*15:01 | 107.01 | | - | GAKEISLSY |
| Non-structural protein 1 | 122–130 | 9 | P3 | HLA-A*02:01 | 23.35 | IEDB_EPITOPE:2022 | AIMEKNIML | - |
| Nuclear export protein | 109–116 | 8 | P3 | HLA-B*18:01 | 16.64 | | VEQEIRTF | VEQEIRTF |
| Nuclear export protein | 111–118 | 8 | P3 | HLA-B*18:01 | 167.48 | | QEIRTFSF | QEIRTFSF |
| Nuclear export protein | 33–40 | 8 | P3 | HLA-B*15:01 | - | | - | TQFESLKL |
| Nuclear export protein | 90–99 | 10 | P3 | HLA-B*18:01 | - | | - | TENSFEQITF |
| Nucleoprotein | 45–52 | 8 | P3 | HLA-B*18:01 | 126.91 | | - | TELKLSDY |
| Nucleoprotein | 404–412 | 9 | P3 | HLA-B*15:01 | 26.29 | | - | GQISIQPTF |
| Nucleoprotein | 450–458 | 9 | P3 | HLA-B*15:01 | 149.35 | | - | SARPEDVSF |
| Polymerase basic protein 2 | 52–61 | 10 | P3 | HLA-A*03:01 | 15.08 | | - | AMKYPITADK |
| RNA polymerase catalytic subunit | 177–185 | 9 | P3 | HLA-B*18:01 | 35.56 | | - | EEMGITTHF |

the individuals (S1 Table). NetMHC [27,28] binding affinity predictions of the observed Wisconsin influenza immunopeptides confirms that they are all likely to be high affinity peptides (< 500 nM) (Fig 3). However, the experimentally observed immunopeptides ranged between 1 and 100 in the rankings, with only Polymerase acidic protein (PA) 10-mer DVVNFVSMEF as the highest ranked prediction. Also of note, although all HLA allotypes have a primary preference for 9-mer peptides, many allotypes have secondary and tertiary epitope length preferences [29]. We observed that influenza immunopeptides were only presented for the secondary length preferences of patient allotypes HLA-A*03:01 and HLA-B*18:01: 10-mers and 8-mers respectively (Fig 3 and Table 1).

Excitingly, from P3 we also identified one class II HLA peptide, derived from the Haemagglutinin protein (Table 1). Although this potential CD4+T cell epitope is not yet proven as functional, it is novel, and as the first such identification in ex-vivo infected tissues, it paves the way for the identification of further CD4-stimulatory peptides. The presentation of the only observed CD4+ T cell epitope derived from a membrane-resident protein is consistent with the predominantly extracellular/membrane origin of HLA-II sourced proteins.

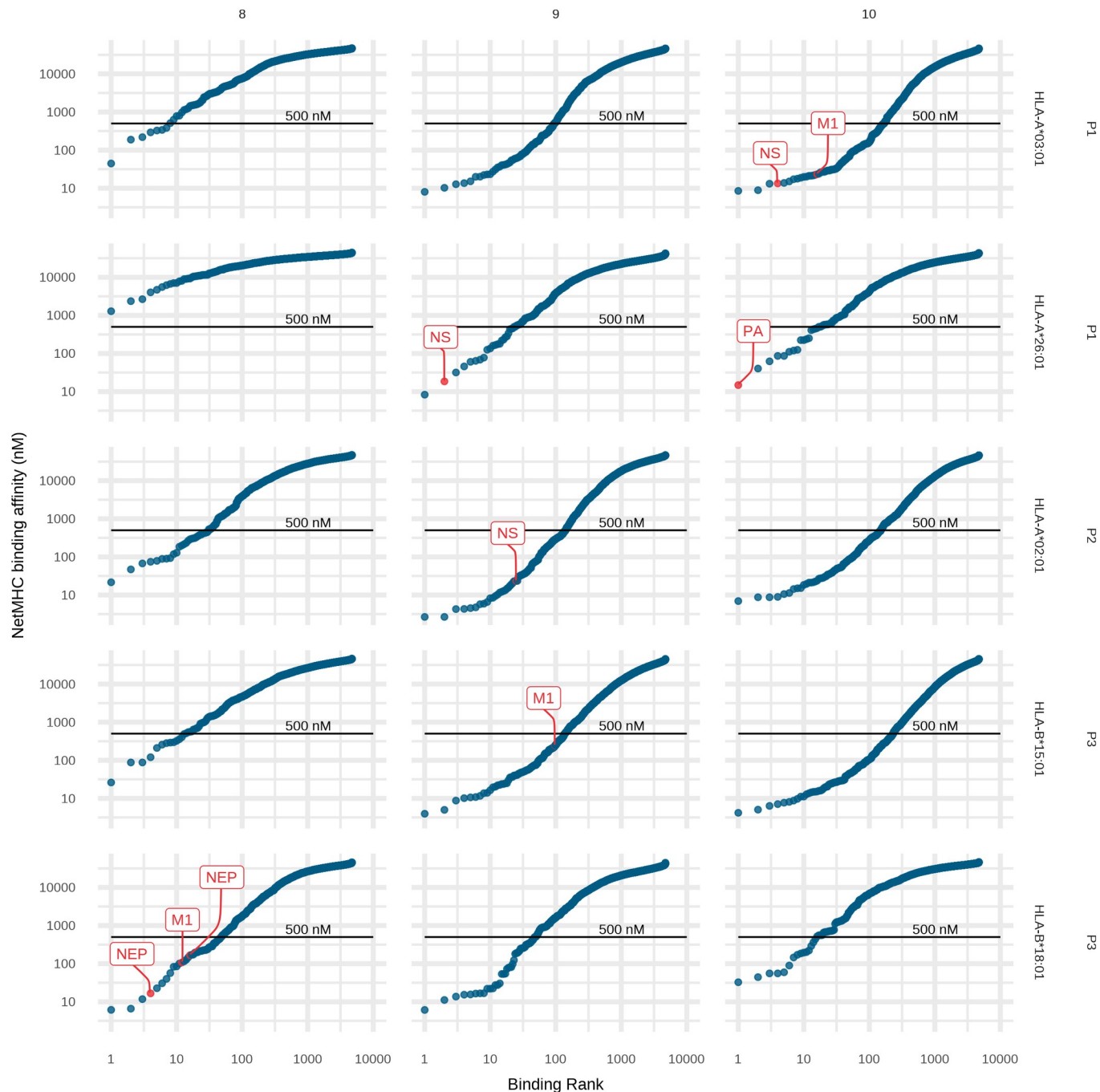

**Fig 3. Wisconsin influenza peptide binding predictions for lung tissue HLA allotypes.** NetMHC binding affinity predictions of the 9 observed Wisconsin influenza immunopeptides and 70,665 possible 8, 9, and 10-mer viral immunopeptides. Binding affinity is plotted against binding rank for the HLA allotype and patient (row) and epitope length (column). Observed peptides and their source protein are indicated in red, with the black line indicating the 500 nM threshold below which a peptide is considered a strong binder.

## Infection of THP1macs leads to selective presentation of HLA-restricted influenza peptides

Flow cytometry of THP1 cells differentiated into a macrophage-like phenotype (THP1MΦ) indicated robust expression of HLA-I but only minimal expression of HLA-II (Fig 4A and 4B).

Following exposure to the Wisconsin H3N2 and X31 influenza strains at a MOI of 1.0, approximately 50% and 90% of the THP1 cells were infected respectively (Figs 4C and S1).

Proteomic analysis of the cell lysates using relative quantitation revealed that the intracellular expression of the different influenza proteins showed a hierarchy of expression where the most abundant proteins were matrix protein 1, nucleoprotein and non-structural protein-1, whereas others, such as polymerase basic proteins 1 and 2 were the least abundant (Fig 4D). Matrix protein 2 and RRBP2 were not detected in the analysis. This pattern was similar between the two influenza strains studied, with slight differences in the proportions of the most abundant proteins. This approximate pattern of expression has been previously observed in purified influenza virions [30,31], however our observation of the relatively high abundance of NP and NS1, similar to that observed in ex-vivo infected lung tissues, may be due to our examination of infected cells rather than virions [32], as these were over-represented in the infection models compared to our initial purified influenza stock (S2 Fig). Notably, the five most abundant proteins are the same as those found to be principal targets for cell mediated immune responses in animal infection models [33].

Immunopeptidomic analysis of eluted HLA-I-bound peptides extracted from THP1MΦ infected with Wisconsin virus resulted in the detection of 10,709 unique peptide sequences matching 3,064 host proteins in the Uniprot human database (S2 Table). Cluster analysis of these peptides indicated the presence of three strong HLA-1 binding motifs (Fig 4E), which were consistent with the HLA types of this cell line (S1 Table). Of the observed peptides, 6,499 could be assigned to the homozygous HLA-A*02:01 and -B*15:11 allotypes on the surface of THP-1 cells on the basis of motif presence. Similarly, infection with X31 resulted in 11,643 unique host peptide sequences derived from 3,308 host proteins in the Uniprot human database (S2 Table).

From the three biological replicates used in the study of THP1MΦ, we detected 9 unique influenza peptides associated with Wisconsin infection and 20 associated with X31 (Table 2). HLA-restricted influenza peptides contained the correct binding motifs for the HLA types of THP1 cells (Fig 4E) [34]. There was only one unique Wisconsin strain peptide, which was derived from PA-X, thus all but one of the peptides found in the Wisconsin strain were from identical regions to those in X31 (with small like-for-like differences in amino acid sequence), and the one unique Wisconsin strain sequence has an identical amino acid sequence in the X31 strain. The additional X31 peptides were potentially due to the higher infection rate of X31 in these cells, leading to more intracellular viral protein. As with the lung tissues, the observed influenza peptides were all predicted to be high affinity peptides but were in the range of 1 to 100 in terms of their affinity ranking (Fig 4F). NetMHC cannot make predictions for peptides with the HLA-B*15:11 motif.

Two of the HLA-A*02:01 immunopeptides, NS1 protein-derived peptide AIMDKNIIL and the M1 peptide RMGAVTTEV have been previously observed following X31 infection in respiratory epithelial cells [35].

The majority of observed viral peptides were assigned to the motif for HLA-B*15:11, whereas the majority of host immune-peptides were predicted to bind to HLA-A*02:01 (Fig 4E and 4F and Table 2). It is unclear whether this is due to preferential tracking of viral proteins to the B allotype, or the presence of favourable HLA-B*15:11 binding motifs in the viral proteins.

Previous MS studies have shown approximately similar numbers of HLA-A and -B immunopeptides in THP1MΦ [36], the reasons for our observation of a greater abundance of HLA-A peptides is unclear, but may reflect technical differences in immunopeptide analysis. There was a notable bias towards presentation of highly conserved internal viral proteins in

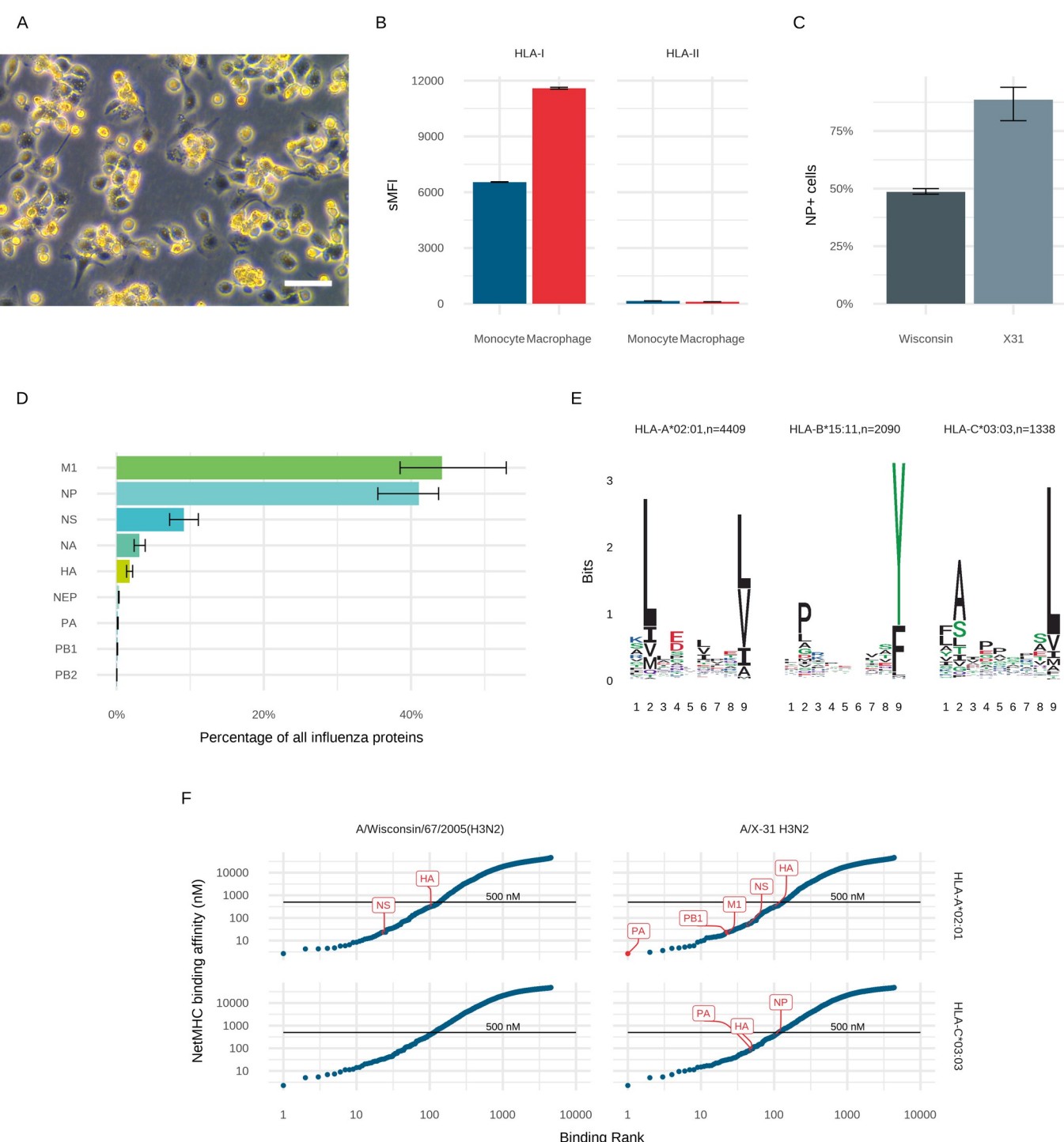

**Fig 4. Characteristics of the THP1MΦ used to identify HLA-binding viral peptides following direct infection with influenza virus.** (**A**) Differentiation of THP1 cells alters cells to a macrophage-like morphology viewed at 20X magnification using phase contrast microscopy. White scale bar, 20 μm (**B**) Differentiation into THP1MΦ increases cell surface HLA-I expression but HLA-II expression remains low. (**C**) Infection rates following exposure to Wisconsin and X31 influenza A strains at an MOI of 1.0. (**D**) Relative proportions of intracellular influenza proteins present in three biological replicates of THP1MΦ (**E**) Class I MHC molecule 9-mer binding motifs for distinct peptides from Wisconsin H3N2 infected THP1MΦ using unsupervised clustering MixMHCp and assigned to HLA types on the basis of known motifs and the known HLA types of these cells. (**F**) NetMHC binding predictions for 9-mer peptides from the influenza virus: 8,884 peptides per strain. Binding affinity is plotted against binding rank for the HLA allotype (row) and influenza strain (column). Observed peptides and their source protein are indicated in red, with the black line indicating the 500 nM threshold below which a peptide is considered a strong binder.

**Table 2. Summary of the HLA-I restricted immunopeptides isolated from THP1 macrophages infected with two influenza A viral strains, their predicted binding allotype and source protein.** Peptides in blue have mis-matched amino-acid sequences between the strains. nM is NetMHC 4.0 predicted binding affinity.

| Protein | Protein Position | Length | Allotype | nM | ID | Wisconsin | X-31 |
|---|---|---|---|---|---|---|---|
| Hemagglutinin | 114–121 | 8 | HLA-B*15:11 | - | | YPYDVPDY | YPYDVPDY |
| Hemagglutinin | 28–36 | 9 | HLA-A*02:01 | 322.25 | | TLCLGHHAV | TLCLGHHAV |
| Hemagglutinin | 310–318 | 9 | HLA-B*15:11 | - | IEDB_EPITOPE:124127 | FQNVNRITY | - |
| Hemagglutinin | 310–318 | 9 | HLA-B*15:11 | - | IEDB_EPITOPE:124126 | - | FQNVNKITY |
| Hemagglutinin | 251–259 | 9 | HLA-C*03:03 | 94.67 | | - | TIVKPGDVL |
| Hemagglutinin | 308–318 | 11 | HLA-B*15:11 | - | | KPFQNVNRITY | KPFQNVNKITY |
| Matrix protein 1 | 232–240 | 9 | HLA-B*15:11 | - | IEDB_EPITOPE:9164 | DLLENLQTY | - |
| Matrix protein 1 | 134–142 | 9 | HLA-A*02:01 | 23.5 | IEDB_EPITOPE:54888 | - | RMGAVTTEV |
| Matrix protein 1 | 107–115 | 9 | HLA-B*15:11 | - | IEDB_EPITOPE:28891 | - | ITFHGAKEI |
| Matrix protein 1 | 232–240 | 9 | HLA-B*15:11 | - | IEDB_EPITOPE:9163 | - | DLLENLQAY |
| Non-structural protein 1 | 122–130 | 9 | HLA-A*02:01 | 23.35 | IEDB_EPITOPE:2022 | AIMEKNIML | - |
| Non-structural protein 1 | 122–130 | 9 | HLA-A*02:01 | 53.78 | IEDB_EPITOPE:2014 | - | AIMDKNIIL |
| Non-structural protein 1 | 68–77 | 10 | HLA-B*15:11 | - | | ILKEESDEAL | ILKEESDEAL |
| Nucleoprotein | 450–458 | 9 | HLA-B*15:11 | - | | - | SARPEDVSF |
| Nucleoprotein | 439–447 | 9 | HLA-B*15:11 | - | | - | DMRTEIIRM |
| Nucleoprotein | 256–264 | 9 | HLA-C*03:03 | 419.04 | IEDB_EPITOPE:768533 | - | LTFLARSAL |
| Protein PA-X | 102–110 | 9 | HLA-B*15:11 | - | | KPKFLPDLY | - |
| Protein PA-X | 27–35 | 9 | HLA-B*15:11 | - | | DLKIETNKF | DLKIETNKF |
| Protein PA-X | 46–54 | 9 | HLA-A*02:01 | 2.66 | IEDB_EPITOPE:17119 | - | FMYSDFHFI |
| Protein PA-X | 46–54 | 9 | HLA-C*03:03 | 94.61 | IEDB_EPITOPE:17119 | - | FMYSDFHFI |
| RNA polymerase catalytic subunit | 162–170 | 9 | HLA-A*02:01 | 23.34 | IEDB_EPITOPE:54584 | - | RLIDFLKDV |
| RNA polymerase catalytic subunit | 22–30 | 9 | HLA-B*15:11 | - | IEDB_EPITOPE:17455 | - | FPYTGDPPY |
| RNA polymerase catalytic subunit | 745–753 | 9 | HLA-B*15:11 | - | | - | KICSTIEEL |
| RNA polymerase catalytic subunit | 22–32 | 11 | HLA-B*15:11 | - | | - | FPYTGDPPYSH |
| RNA polymerase catalytic subunit | 28–38 | 11 | HLA-B*15:11 | - | | - | PPYSHGTGTGY |

the HLA-I peptidome, with only four nested haemagglutinin peptides and no neuraminidase peptides detected in the infected cell line.

The majority of our observed peptides in THP1MΦ have been previously characterised by in-vitro binding/cytotoxicity assays and were present in the Immune Epitope Database (IEDB) (Tables 2 and S4), although not derived from the two strains studied herein. Most reported positive ELISpot outputs, confirming that they led to functional responses. The majority of these immunogenic peptides were previously identified because many influenza strains have been intensively studied.

With the exception of the well-known HLA-A*02:01 peptide AIMEKNIML/AIMDKNIML, and the HLA-B*15:01/HLA-B*15:11 peptide SARPEDVSF, there was no overlap in the detected influenza peptides between the lung tissues samples and THP1MΦ due to the diverse nature of the HLA allotypes in these randomly selected lung tissue samples.

Very few host cell HLA-II peptides could be detected on these cells, consistent with our flow cytometry data, suggesting that expression of HLA-II on the cell surface was low (Fig 4B). No influenza peptides were detected bound to HLA-II.

We found some influenza immunopeptides from proteins which were undetectable in the proteome of infected cells. Such a finding is consistent with previous reports that immunopeptide selection is poorly correlated with source protein concentration [37], but may also reflect the challenges of detecting lower abundance proteins in a complex proteome such as that derived from infected cell lines.

## Phagocytosis of apoptotic influenza-infected MoDCs reveals multiple nested MHC-II influenza epitopes

Using Wisconsin H3N2-infected A549 cells (80% infected, see S3 Fig) as the source of viral proteins, we UV irradiated these cells to drive them into apoptosis prior to feeding them to MoDCs from a heterotypic HLA type individual (Fig 5A; Patient ID P4).

Phagocytosis of these infected cells led to preferential presentation of HLA-II bound influenza peptides (Table 3). Those peptides containing motifs matching the HLA-II allotypes (S1 Table, P4), with no observable viral HLA-I peptides, despite robust host-derived HLA-I peptide presentation in these cells (S2 Table). This lack of evidence of cross-presentation of influenza HLA-I peptides by human DCs has been previously observed when using an HLA-A*02 cell line (BEAS-2B) [35].

We observed 4,639 distinct class II HLA peptides deriving from 891 source proteins (S2 Table). Motif deconvolution [38] was able to assign 2,597 peptides to the respective HLA-DRB1 allotypes of P4 (Fig 5B). Within these immunopeptidomes, there were 29 influenza A derived HLA-II restricted peptide sequences. Contrary to viral presentation following direct infection of cells or tissues, there was a very strong bias to the processing and display of the membrane-bound proteins neuraminidase, haemagglutinin, and Matrix protein 1 in the detected HLA-II peptides (Table 3).

Binding predictions for HLA-DRB1 using NetMHCIIpan [39] which can only predict 15-mers, indicated that the observed peptides were high affinity, but not top-ranking predictions (Fig 5C). However, multiple nested peptides were detected for the majority of these viral HLA-II epitopes, similar to those for the majority of host proteins, suggestive of permissive regions of these viral proteins for processing and presentation.

For example, three motifs corresponding to the HLA-DRB1 allotype could be identified in the M1 protein. When overlaid onto the M1 protein sequence [40], these HLA-II motifs were located predominantly towards the C terminus of the amino acid sequence, whereas the HLA-I motifs were equally distributed over the protein, including regions also presented by HLA-II (Fig 5D), possibly reflective of the different processing pathways involved in HLA-I and -II presentation.

Previous work has shown that, when pulsed with recombinant influenza haemagglutinin, MoDCs will present HLA-II immunopeptides from select regions of the protein corresponding to the immune-dominant memory T cell population with higher avidity than naïve T cells [41]. Although we did not observe many haemagglutinin HLA-II peptides, we observed a similar pattern of multiple nested peptides from selected regions within the M1 protein. Cryo-EM has recently revealed the assembled structure of the M1 protein as multiple helical arrays that polymerise to form the viral endoskeleton, an assembly which unravels in low pH triggering disassembly of the M1 assembly in the endosome [42].

The studies on Haemagglutinin examined endocytosis of recombinant protein, identifying that the head portion of the protein results in the major T cell dominant clones, but our method using phagocytosis of infected respiratory cells renders the majority of viral proteins subject to antigen processing by APCs, potentially resulting in a wider range of immunodominant HLA-II peptides. The reason for the predominance of M1 protein in these HLA-II peptides is not clear but could reflect the high intracellular abundance of this protein.

Analysis of A549 cells infected with the Wisconsin strain identified only three HLA-I restricted peptides, and no HLA-II restricted peptides (S3 Table). All were consistent with the HLA-I allotypes of these cells. We found no evidence of either HLA-I or–II peptides sourced from the A549 cells (i.e. matching their allotypes) following engulfment by DCs. This was perhaps not surprising considering how few influenza peptides appear to be presented by these

A

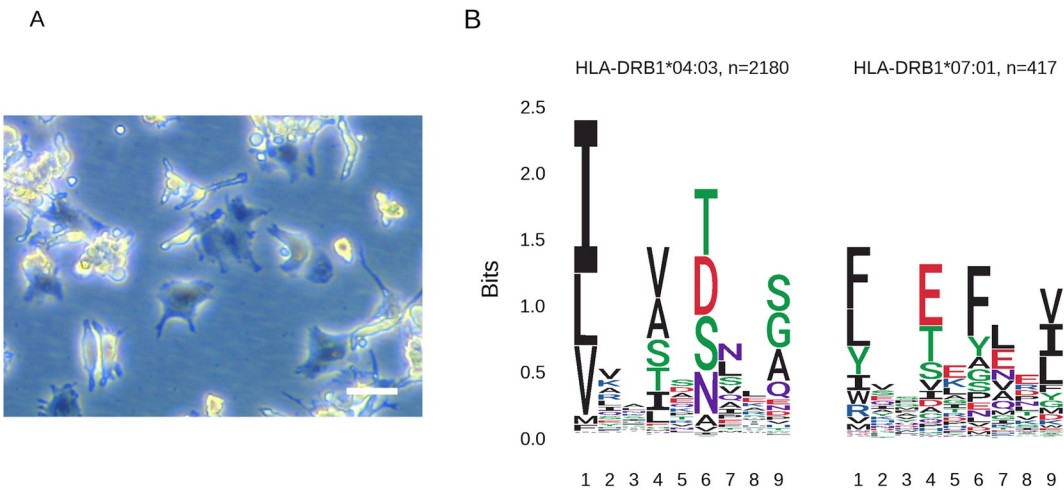

D

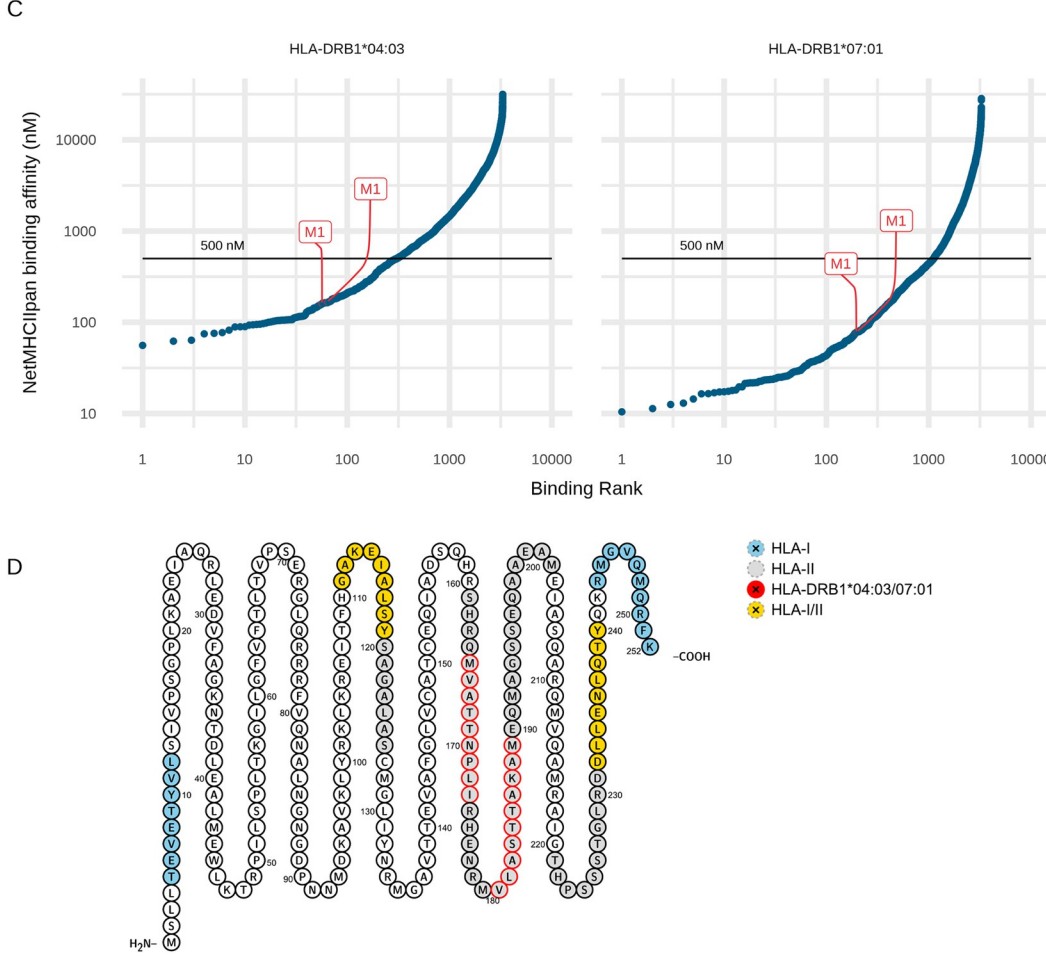

**Fig 5. MoDCs can be used to identify HLA-II binding immunopeptides following phagocytosis of apoptotic influenza-infected respiratory epithelial (A549) cells.** (A) Dendritic cell morphology prior to exposure to apoptotic A549 cells, 20X magnification using phase contrast microscopy. White scale bar, 50 μm. (B) Motif deconvolution using MoDec yields class II HLA-DRB1 molecule core binding motif logos for distinct peptides presented by MoDCs following phagocytosis of A549 cells infected with Wisconsin H3N2. (C) NetMHCIIpan predictions of 6,581 15mer Wisconsin influenza peptides. Binding affinity is

plotted against binding rank for the HLA allotype (row) and influenza strain (column). Observed peptides and their source protein are indicated in red, with the black line indicating the 500 nM threshold below which a peptide is considered a strong binder. **(D)** The sequence of Matrix protein 1 and all corresponding observed influenza peptides in our study: regions generating class I and II epitopes are in yellow, class I only observed epitopes in blue, and class II only observed epitopes in grey. The predicted HLA-DRB1 motifs are highlighted with red circles.

cells, and the fact that only $10^6$ cells were used in the MoDC assay (only 10% of the amount normally required to achieve a peptidome of >1,000 unique peptides).

## Discussion

Current subunit vaccine strategies to optimize T cell responses to influenza challenge are mostly directed towards the most mutable proteins such as Haemagglutinin and Neuraminidase. There is evidence to suggest that HLA-I restricted T cell responses are more directed towards the more highly conserved internal viral proteins, whereas humoral responses are dominated by envelope proteins [43]. Despite the proteome of influenza comprising of only a dozen proteins, this yields many thousands of potential T cell epitopes. Therefore, identifying the epitopes most important for anti-influenza responses by predictive means is challenging. Most viral proteins will contain HLA binding motifs for multiple allotypes, but current evidence suggests that only a small minority of these will actually be presented [44].

Here we show how influenza epitope presentation is influenced by presence of HLA binding motifs, source protein abundance, and the HLA pathway. We confirm that only a few internal viral proteins provide the main source of HLA-I immunopeptides in lung tissues, and we find that select immunopeptides are favoured in different influenza strains. Viral protein abundance influences but is not the only factor in HLA presentation. Using a MoDC model we show that viral membrane bound proteins such as NA, HA and M1 are preferentially presented by HLA-II, and that certain regions of these proteins may be more conducive to processing via the HLA-II pathway. These results demonstrate how peptidomics can reduce the potential pool of anti-influenza T cell epitopes from thousands to a few dozen. Furthermore, these candidates can be refined according to their relevant HLA pathway and helps guide predictive algorithm epitope selection more effectively.

To address the issue of viral tropism, we have taken an approach using ex-vivo human lung tissues as the targets for infection with different strains of influenza, one model strain (A/H3N2/X31) and one circulating strain from 2005 (A/H3N2/Wisconsin/2005/67).

Our work has previously demonstrated that these strains are capable of infecting both epithelial cells and macrophages in ex-vivo lung tissues and inducing disease-relevant inflammatory responses which can be modulated using either anti-viral or anti-inflammatory medications [23]. We predicted that due to the presence of significant numbers of alveolar macrophages with high intrinsic HLA-DR expression, we would simultaneously generate HLA-I and–II peptides using this lung tissue model. However, despite demonstrable infection of both epithelial cells and macrophages in our experiments and identifying a number of novel HLA-I epitopes in these tissues with relative ease, we were only able to identify one HLA-II peptide. This observation may help to substantiate previous evidence that the primary function of alveolar macrophages is to kill and phagocytose infected cells [45], and that DCs are more efficient at presenting HLA-II peptides than macrophages, whether this is by direct infection or following phagocytosis of infected cells.

CD8+ T cell responses are recognised as key components of the adaptive immune response to viral infections. Since they are mostly directed against conserved internal proteins of the influenza virus, they are thought to provide significant cross-protection between different strains. For this

**Table 3. Summary of HLA-II restricted immunopeptides isolated from DCs following engulfment of influenza-infected A549 cells.** nM is NetMHC 4.0 predicted binding affinity.

| Peptide | Protein | Protein Position | Length | Motif | Allotype | nM | ID |
|---|---|---|---|---|---|---|---|
| LNNRFQIKGVELK | Hemagglutinin | 512–524 | 13 | - | Unassigned | - | |
| ALNNRFQIKGVELK | Hemagglutinin | 511–524 | 14 | - | Unassigned | - | |
| GAKEIALSYSAGAL | Matrix protein 1 | 111–124 | 14 | - | Unassigned | - | |
| PSGPLKAEIAQRLE | Matrix protein 1 | 16–29 | 14 | - | Unassigned | - | |
| ENRMVLASTTAKAME | Matrix protein 1 | 176–190 | 15 | VLASTTAKA\|LASTTAKAM | HLA-DRB1*04:03/07:01 | 161.63 | IEDB_EPITOPE:13582 |
| ENRMVLASTTAKAME | Matrix protein 1 | 176–190 | 15 | VLASTTAKA\|LASTTAKAM | HLA-DRB1*04:03/07:01 | 78.62 | IEDB_EPITOPE:13582 |
| GAKEIALSYSAGALA | Matrix protein 1 | 111–125 | 15 | - | Unassigned | 2455.17 | |
| NRMVLASTTAKAMEQ | Matrix protein 1 | 177–191 | 15 | VLASTTAKA\|LASTTAKAM | HLA-DRB1*04:03/07:01 | 166.75 | IEDB_EPITOPE:1309539 |
| NRMVLASTTAKAMEQ | Matrix protein 1 | 177–191 | 15 | VLASTTAKA\|LASTTAKAM | HLA-DRB1*04:03/07:01 | 77.86 | IEDB_EPITOPE:1309539 |
| ENRMVLASTTAKAMEQ | Matrix protein 1 | 176–191 | 16 | VLASTTAKA\|LASTTAKAM | HLA-DRB1*04:03/07:01 | - | |
| GAKEIALSYSAGALAS | Matrix protein 1 | 111–126 | 16 | - | Unassigned | - | |
| HENRMVLASTTAKAME | Matrix protein 1 | 175–190 | 16 | VLASTTAKA\|LASTTAKAM | HLA-DRB1*04:03/07:01 | - | |
| ENRMVLASTTAKAMEQM | Matrix protein 1 | 176–192 | 17 | VLASTTAKA\|LASTTAKAM | HLA-DRB1*04:03/07:01 | - | |
| HENRMVLASTTAKAMEQ | Matrix protein 1 | 175–191 | 17 | VLASTTAKA\|LASTTAKAM | HLA-DRB1*04:03/07:01 | - | IEDB_EPITOPE:128850 |
| HGAKEIALSYSAGALAS | Matrix protein 1 | 110–126 | 17 | - | Unassigned | - | |
| SHRQMVATTNPLIRHEN | Matrix protein 1 | 161–177 | 17 | MVATTNPLI | HLA-DRB1*07:01 | - | |
| HENRMVLASTTAKAMEQM | Matrix protein 1 | 175–192 | 18 | VLASTTAKA\|LASTTAKAM | HLA-DRB1*04:03/07:01 | - | IEDB_EPITOPE:606271 |
| HENRMVLASTTAKAMEQMA | Matrix protein 1 | 175–193 | 19 | VLASTTAKA\|LASTTAKAM | HLA-DRB1*04:03/07:01 | - | |
| HENRMVLASTTAKAMEQMAG | Matrix protein 1 | 175–194 | 20 | VLASTTAKA\|LASTTAKAM | HLA-DRB1*04:03/07:01 | - | |
| QMVQAMRAIGTHPSSSTGLR | Matrix protein 1 | 211–230 | 20 | - | Unassigned | - | |
| HENRMVLASTTAKAMEQMAGSSEQ | Matrix protein 1 | 175–198 | 24 | VLASTTAKA\|LASTTAKAM | HLA-DRB1*04:03/07:01 | - | |
| HENRMVLASTTAKAMEQMAGSSEQAA | Matrix protein 1 | 175–200 | 26 | VLASTTAKA\|LASTTAKAM | HLA-DRB1*04:03/07:01 | - | |
| HENRMVLASTTAKAMEQMAGSSEQAAE | Matrix protein 1 | 175–201 | 27 | VLASTTAKA\|LASTTAKAM | HLA-DRB1*04:03/07:01 | - | |
| HENRMVLASTTAKAMEQMAGSSEQAAEA | Matrix protein 1 | 175–202 | 28 | VLASTTAKA\|LASTTAKAM | HLA-DRB1*04:03/07:01 | - | |
| HENRMVLASTTAKAMEQMAGSSEQAAEAM | Matrix protein 1 | 175–203 | 29 | VLASTTAKA\|LASTTAKAM | HLA-DRB1*04:03/07:01 | - | |
| QMVQAMRAIGTHPSSSTGLRDDLLENLQTY | Matrix protein 1 | 211–240 | 30 | - | Unassigned | - | |
| IEEGKIVHTSTLSGSAQ | Neuraminidase | 257–273 | 17 | IVHTSTLSG | HLA-DRB1*04:03 | - | |
| IEEGKIVHTSTLSGSAQH | Neuraminidase | 257–274 | 18 | IVHTSTLSG | HLA-DRB1*04:03 | - | |
| SPRGKLSTRGVQIASN | Nucleoprotein | 353–368 | 16 | - | Unassigned | - | |

reason, vaccines designed to promote T cell protection against conserved T cell epitopes of multiple IFV strains are highly desirable. The diversity of HLA types in the human population has created challenges in the generation of universal T cell vaccines, as the selected optimal T cell epitopes must reflect the HLA restriction of the target population as much as possible.

To better understand the HLA-I viral immunopeptidome, we initially used a cell model (THP1) with the aim of identifying T cell epitopes for influenza. These cells were more susceptible to infection with the laboratory-adapted X31 strain than the more clinically relevant Wisconsin strain. We were able to identify a number of well-characterised HLA-A and B epitopes that had been previously observed in similar studies. The utility of this approach is limited by the molecular phenotype of THP1 cells which are homozygous for the three indicated haplotypes. Bioinformatic comparison of our observed epitopes with a predicted list found that most of the HLA-A and B epitopes detected using this cell model were predicted binders for the known THP1 allotypes, but only represented a small proportion of the predicted binders. The reasons for this are complex, and do not necessarily imply that others are not present, but rather that they may be unrecognised, since MS is biased to the detection of peptides with certain biophysical characteristics. For example, the well-characterised immunodominant M158-66 peptide GILGFVFTL, is a high-ranking predicted HLA-A*02:01 immunopeptide which is refractory to identification by mass spectrometry.

A number of the T cell epitopes we identified were derived from the identical region of the same protein in the two influenza strains despite small differences in amino acid sequence. This may arise because the HLA anchor positions were not altered, but is also suggestive of intrinsic properties of these protein regions being conducive to antigen processing and presentation.

Some of our observed influenza immunopeptides did not match with any allotype or were assigned to HLA-C, but with low predicted affinity. This may reflect the poor performance for predictive algorithms using the C allotype which is less well characterised. Often where the motifs for HLA binding are not clearly defined, prediction tools are less useful, meaning direct observation could play a more significant role, not only in identifying novel peptides, but also in improving the algorithms for future searches. HLA-II prediction algorithms are thought to be even less reliable [46].

Despite differentiation of THP1 cells into a macrophage-like phenotype, we did not generate significant numbers of host-cell HLA-II ligands, and saw no evidence of influenza HLA-II ligands, although the identified HLA-I binding viral peptides were consistent between biological experimental replicates confirming the robustness of our immunopeptidome isolation methodology. Recent work on tuberculosis using the THP1 cell line treated with a cytokine mixture designed to increase HLA-II expression identified some HLA-II epitopes [47], and a similar approach could be used for influenza in the future.

There are currently few reliable in-vitro cell culture methods of identifying HLA-II epitopes for any virus. These methods have the further limitation of being useful only for laboratory adapted strains of influenza which are capable of productive infection in cell lines, and can only be used to identify T cell epitopes for the HLA type of the cell line used.

To improve and increase the efficiency of identifying naturally processed and presented HLA-II epitopes, we employed a human infection model, where we infected a heterotypic monolayer cell line (A549) with influenza virus to generate intracellular virus particles, and then drove those cells into apoptosis using UV irradiation to facilitate phagocytosis by in-vitro derived dendritic cells. This methodology resulted in robust generation of HLA-II epitopes of the outer coat proteins of influenza virus, in addition to matrix 1 proteins, but not of the internal core proteins of the virus. No evidence of influenza HLA-II peptides matching the HLA-types of the A549 cells were observed. Neither were viral HLA-I peptides generated using this method, suggestive of a lack of HLA-I cross-presentation, at least when using this influenza

strain. This is the first time that candidate influenza HLA-II epitopes have been directly observed in a fully human infection model capable of generating personalised CD4+ T cell epitopes. Whilst we used a cell line infected with virus as the source material which could be susceptible to strain tropism issues, it would be entirely feasible to use any cells as the source material since their HLA type would not be relevant to the epitopes discovered, and very few cells express significant amounts of HLA-II apart from professional APCs. HLA-II help is essential for an effective vaccine, since although non-specific CD4+ help can promote CD8+ T cell effector functions, specific CD4+ T cells are required for the proper generation of memory CD4+ cells [48]. Recent evidence also suggests that CD4 cells reacting to peptides from core viral proteins such as NP and M1 are first responders to influenza challenge and can have important effector functions of their own, as they contain perforin and granzyme and produce IFN-γ [14]. Furthermore, the requirement for CD4 T cell epitopes to facilitate CD8 T cell killing in addition to humoral responses means that the direct observation of HLA-II peptides will be extremely valuable in improving vaccines for combating potential influenza outbreaks in the future.

Live vaccines confer more effective CD8+ T cell responses than attenuated ones which stimulate mostly CD4+ T cell responses [49]. The DC uptake model indicates strong compartmentalisation of CD4 and CD8 processing and peptide display and indicates that a combination of uptake and direct infection is likely to be most effective in vaccine generation.

It has been proposed that alternative pathways of antigen processing in infected APCs, rather than virion or infected cell uptake, is the primary driver of CD4+ T cell response to influenza infection [50]. Further work might reveal differential immunopeptidomes in MoDCs infected directly with influenza virus, rather than following uptake of infected cells as in our study. Here we have demonstrated the potential for fully human ex-vivo models as tools to identify viral immunopeptides which could be used to design strain-specific T cell vaccines against influenza. We have shown evidence that these epitopes will be conserved between different donors if they share the same HLA allotype.

Much of the vaccine design process employs machine learning algorithms to predict the relevant HLA-I CD8 T cell epitopes by searching for motifs and predicting their affinity in-silico. There is increasing evidence, including in our data, that epitopes that are actually presented are influenced in-vivo by a complex series of additional factors however, inclusive of source protein abundance, protease type in the proteasome, protein turnover, transporter protein expression, PTMs and many others, making the prediction algorithms liable to prioritise non-immunogenic peptides.

There are many potential combinations of HLA allotypes in humans, and the best T cell epitopes need to be selected. The current approach is limited by the quality of HLA-I and–II epitopes that can be predicted using the current algorithms. These algorithms are trained using mass spectrometry (MS) data from peptides eluted from the HLA complex, and thus, rarer HLA types often have less well-defined motifs, and the predictions are therefore less accurate. In the case of HLA-II this is even more apparent, as the paucity of eluted peptide datasets for each haplotype means the predictions are probably highly inaccurate. Furthermore, model cell lines may not be susceptible to pathogen infections.

Using our ex-vivo lung tissue and DC infection models, a panel of HLA-I and–II T cell epitopes for a known pathogen could be used to generate a vaccine based on real-world observations of influenza A virus immunopeptides for a range of HLA types, potentially enhancing vaccine efficacy.

## Materials and methods

### Ethics statement

Informed written consent was provided for participation by all individuals. This study was approved by the University of Southampton Research Ethics Committee.

## Virus propagation

A/H3N2/X31 and A/H3N2/Wisconsin/67/2005 seed stocks were obtained from the National Institute for Biological Standards and Control (NIBSC), UK, propagated in embryonated SPF-free chicken eggs and, subsequently, purified from egg allantoic fluid by sucrose density gradient ultracentrifugation (Virapur LLC, San Diego, USA). The X31 virus contains the six internal genes of PR8, an H1N1 laboratory-adapted influenza virus strain, but expresses H3N2 surface proteins. Stock viral titre was determined by MDCK plaque assay using standard protocols. To generate inactivated UV virus, aliquots were irradiated for 30 min on ice using an ultraviolet microbicidal crosslinker (Steristrom 2537a) as previously described [51].

## Cell culture

The acute myeloid leukaemia cell line THP1 was cultured in RPMI supplemented with 10% FCS and 1% penicillin-streptomycin. Cells ($1.5 \times 10^8$ per treatment arm) were differentiated into macrophage-like cells (THP1MΦ) by incubation with 100 ng/mL Phorbol-12-myristate-13-acetate (PMA) for 48h in complete medium followed by a 24 h rest period in complete medium without PMA, by which time the majority of cells were adherent to the culture surface.

A549 cells were cultured in DMEM supplemented with 10% FCS and 1% penicillin-streptomycin and passaged at approximately 80% confluence.

## Cell infections

Cell monolayers were rinsed three times with basal medium to remove serum proteins before 2 h infection with either A/H3N2/Wisconsin/67/2005 or A/H3N2/X31 influenza virus in basal medium at the indicated multiplicity of infection (MOI), followed by a further incubation as specified. Mock infections with UV-inactivated virus were performed at the same dose. Following infection, cell monolayers were rinsed twice with PBS and the cells were harvested by trypsin treatment, washed twice with 50 volumes of PBS prior to storage as a cell pellet at -80˚C. A small aliquot of cells was preserved for flow cytometry.

## Human lung tissue explants

Resected human lung parenchymal tissues from three donors undergoing surgery for clinical reasons were obtained. Parenchymal tissue without evidence of visible abnormalities, distant from the resection margin (250 mg of fresh tissue per treatment), was dissected and placed into culture within 2 h. The study was performed in accordance with Research Ethics Committee (REC) approvals, (Southampton and South West Hampshire Research Ethics Committee, LREC no: 09/H0504/109). Parenchymal explants were first rested overnight in RPMI in 24-well culture in a humidified incubator at 37˚C with 5% CO2, then cultured in glutamine-supplemented RPMI medium with log 7.4 pfu/well virus, either A/H3N2/X31 or A/H3N2/Wisconsin/67/2005 (National Institute for Biological Standards and Control, UK or virus diluent as mock control. After 2 h incubation, the explants were washed in basal RPMI medium to remove excess virus and incubated further for 22 hours in glutamine-supplemented RPMI. At the end of the incubation, tissues were washed with 50 volumes of PBS and snap frozen as pellets prior to storage at -80˚C. Approximately 20 mg of fresh tissue were removed prior to freezing for flow cytometric analysis.

## Flow cytometry

**Assessment of infection in A549 cells and basal and differentiated THP-1 cells.** For analysis of influenza infection, monolayer cells were harvested and re-suspended in 200 μL of

Fixation/permeabilisation solution (BD biosciences, San Jose, CA), and incubated on ice for 30 min. The cells were then harvested by centrifugation (400g, 4°C, 5 min) and re-suspended in 100 μL permeabilisation buffer (FACS buffer, 1% BSA, 1 mM EDTA in PBS containing 1 X BD permeabilisation reagent). FITC-conjugated anti-influenza NP monoclonal antibody (Abcam clone 20921) was added (1 μL per reaction) and incubated for 30 min on ice. Excess antibody was then removed by the addition of 2 mL of permeabilisation buffer followed by centrifugation as before. Cells were finally re-suspended in FACS buffer without permeabilisation reagents prior to flow cytometric analysis using either a BD FACSAria or a Guava EasyCyt flow cytometer, equipped with appropriate lasers and filters.

**Analysis of MHC expression on differentiated THP1 cells.** THP1MΦ were harvested by centrifugation and re-suspended in 100 μL FACS buffer containing 10% FCS. 1 μg/mL anti-HLA-I (W6/32) or anti-HLA-2 (HB-145) monoclonal antibodies were added and incubated for 30 min on ice. Excess antibodies were removed by centrifugation as previously described and cells were re-suspended in 100 μL FACS buffer containing 1 μL per reaction of FITC conjugated rabbit anti-mouse monoclonal antibodies and incubated for a further 30 min on ice. Cells were centrifuged as before and re-suspended in 200 μL of 2% (w/v) formalin and incubated on ice for 30 min to fix the cells. Finally, cells were re-suspended in FACS buffer prior to analysis using a GUAVA EasyCyte benchtop flow cytometer (Merck Millipore) equipped with the relevant laser and filters to detect FITC fluorescence. Data were analysed using GUAVA software. HLA-I and–II FMO were calculated by gating against cells incubated with secondary antibody alone.

**Assessment of influenza infection in resected human lung tissue samples.** To analyse influenza infection in resected tissue samples, post-infection, tissues were weighed and enzymatically dispersed with 1 mg/mL type I collagenase in RPMI as previously described 24. Dispersed cells were re-suspended in 100 μL FACS buffer containing human IgG (as Fcγblock) prior to the addition of antibodies directed against surface proteins: CD45-PECF594 (to differentiate leukocytes from structural cells), CD3-PECy7, HLA-DR/APCH7 and CD326-PerCPCy5.5 or relevant fluorophore-conjugated isotype controls, and incubated for 30 min on ice. Cells were then fixed and permeabilised as previously described, prior to intracellular staining to quantify viral infection using FITC-conjugated anti-viral nucleoprotein (NP) antibody. All flow cytometry was performed using a BD FACSAria equipped with the relevant lasers and filters, and data were analysed using BD FACS DIVA software. Epithelial cells were identified using the following gating strategy: size/scatter, CD45-, CD326+. Macrophages were identified using the following gating strategy: size/scatter, CD45+, CD3-, HLA-DR++ (S4 Fig). Infected epithelial and macrophage cell populations were identified by NP-FITC staining, gated against mock-infected controls using a 1% overlap (S4E and S4F Fig).

## Phagocytosis assay of MoDCs to identify HLA-II viral ligands

**Isolation and preparation of MoDCs.** PBMCs were isolated from buffy coats and allowed to adhere to tissue culture treated flasks for 2 h in monocyte attachment buffer (Product: C-28051, PromoCell GmbH, Germany). Monolayers were then rinsed thoroughly with DC generation medium (Product: C-28050, PromoCell GmbH, Germany) to remove non-adherent cells and subsequently were cultured for 6 days in dendritic cell generation medium supplemented with cytokines (Product: C-28050, PromoCell GmbH, Germany) to generate immature MoDCs.

**A549 cell infection and apoptosis.** Monolayer cultures of A549 cells were infected and sent into apoptosis essentially as previously described [52]. Briefly, 90% confluent monolayers of A549 cells were infected with A/Wisconsin/67/2005 influenza at an MOI of 1.0 in serum-

free medium for 2 h, monolayers were rinsed twice with serum free medium to remove excess inoculum, and then cultured for a further 12 h in serum-free DMEM supplemented with penicillin-streptomycin and l-glutamine for a further 12 h. The infection rate of >80% was confirmed by flow cytometry (S3 Fig) using detection of intracellular viral NP protein as previously described. The infected cell monolayers were rinsed twice with PBS and then irradiated with 150 mJ/cm$^2$ of UV light using a Stratalinker 1800 (Agilent technologies, Santa Cruz, CA, USA) to induce apoptosis. The cells were then incubated for a further 2 h in serum-free medium, prior to enzymatic dispersal to collect the cells. These were then re-suspended at a concentration of 1x10$^7$/mL in monocyte generation medium and added to the MoDCs for 3 h. DC activation supplement was then applied and the cells incubated for a further 4 h. Cells were then harvested by trypsinisation and washed twice by centrifugation in PBS before storage at -80˚C. MoDCs generated using the PromoCell DC kit and subsequently stimulated with the supplied activation supplement exhibit a CD14- / CD45+ / CD83+ phenotype.

## Immunopeptidome analysis

**Purification HLA-I and–II immunopeptides.** Protein-A sepharose beads (Repligen, Waltham, Mass. USA) were covalently conjugated to 10 mg/mL W6/32 (pan-anti-HLA-I) or 5 mg/mL HB145 (pan-anti-HLA-II) monoclonal antibodies (SAL Scientific, Hampshire, UK) using DMP as previously described [53]. Snap frozen tissue samples were briefly thawed and weighed prior to 30 S of mechanical homogenization using a 150W handheld mechanical homogeniser with disposable probes (Thermo Fisher Scientific) in 4 mL lysis buffer (0.02M Tris, 0.5% (w/v) IGEPAL, 0.25% (w/v) sodium deoxycholate, 0.15mM NaCl, 1mM EDTA, 0.2mM iodoacetamide supplemented with EDTA-free protease inhibitor mix). For cell cultures, frozen cell pellets were re-suspended in 5 mL of lysis buffer and rotated on ice for 30 min to solubilise.

Homogenates were clarified for 10 min @2,000g, 4˚C and then for a further 60 min @13,500g, 4˚C. 2 mg of anti-HLA-I conjugated beads were added to the clarified supernatants and incubated with constant agitation for 2 h at 4˚C. The captured HLA-I/β$_2$microglobulin/ immunopeptide complex on the beads was washed sequentially with 10 column volumes of low (isotonic, 0.15M NaCl) and high (hypertonic, 0.4M NaCl) TBS washes prior to elution in 10% acetic acid and dried under vacuum. The MHC-I-depleted lysate was then incubated with 1 mg of anti-HLA-II mouse monoclonal antibodies and MHC-II bound peptides were captured and eluted in the same conditions. Column eluates were diluted with 0.5 volumes of 0.1% TFA and then applied to HLB-prime reverse phase columns (Waters, 30 mg sorbent/column). The columns were rinsed with 10 column volumes of 0.1% TFA and then the peptides were eluted with 12 sequential step-wise increases in acetonitrile from 2.5–30%. Alternate eluates were pooled and dried using a centrifugal evaporator and re-suspended in 0.1% formic acid.

**LC-MS/MS analysis of HLA-I and -II peptides.** HLA peptides were separated by an Ultimate 3000 RSLC nano system (Thermo Scientific) using a PepMap C18 EASY-Spray LC column, 2 μm particle size, 75 μm x 50 cm column (Thermo Scientific) in buffer A (0.1% Formic acid) and coupled on-line to an Orbitrap Fusion Tribrid Mass Spectrometer (Thermo Fisher Scientific, UK) with a nano-electrospray ion source. Peptides were eluted with a linear gradient of 3%-30% buffer B (Acetonitrile and 0.1% Formic acid) at a flow rate of 300 nL/min over 110 minutes. Full scans were acquired in the Orbitrap analyser using the Top Speed data dependent mode, performing a MS scan every 3 second cycle, followed by higher energy collision-induced dissociation (HCD) MS/MS scans. MS spectra were acquired at resolution of 120,000 at 300 m/z, RF lens 60% and an automatic gain control (AGC) ion target value of 4.0e5 for a

maximum of 100 ms. MS/MS resolution was 30,000 at 100 m/z. Higher-energy collisional dissociation (HCD) fragmentation was induced at an energy setting of 28 for peptides with a charge state of 2–4, while singly charged peptides were fragmented at an energy setting of 32 at lower priority. Fragments were analysed in the Orbitrap at 30,000 resolution. Fragmented m/z values were dynamically excluded for 30 seconds.

**Data analysis for immunopeptidomics.** Raw spectrum files were analysed using Peaks Studio 10.0 build 20190129, with the data processed to generate reduced charge state and deisotoped precursor and associated product ion peak lists which were searched against a Uniprot database (20,350 entries, 2020–04) appended with the full sequences for both influenza strains: A/Wisconsin/67/2005(H3N2), 12 entries or A/X-31(H3N2), 11 entries. A contaminants list (245 entries) in unspecific digest mode was applied. Parent mass error tolerance was set a 5ppm and fragment mass error tolerance at 0.03 Da. Variable modifications were set for N-term Acetylation (42.01 Da), Methionine oxidation (15.99 Da) and carboxyamidomethylation (57.02 Da) of cysteine. A maximum of three variable modifications per peptide were set. The false discovery rate (FDR) was estimated with decoy-fusion database searches and were filtered to 1% FDR. The search results were further refined using the MS-Rescue package [54]. Downstream analysis and visualizations were performed in R using associated packages [26,55–57]. Peptide binding motifs were identified using unsupervised clustering methods MixMHCp2.1 [25] and MoDec [38], for class I and class II HLA peptides respectively. Peptide binding affinities predicted using NetMHC 4.0 [27,28] and NetMHCIIpan 4.0 [39] for class I and class II HLA peptides respectively.

## Proteomic profiling

**Sample preparation.** 100 μg of protein from HLA-I and–II depleted cell and tissue lysate were precipitated using methanol/chloroform extraction. Lysate containing 100 μg of protein were mixed with 600 μL of methanol and 150 μL of chloroform. 450 μL of water were added and the sample was centrifuged at 13,500 g for 10 min at room temperature. The upper aqueous later was removed and replaced with 450 μL of methanol and the sample centrifuged again to pellet the proteins. The protein pellet briefly air-dried prior to resuspension in 100 μL of 6M urea/50 mM Tris-HCl pH 7.4. The sample was sequentially reduced and alkylated by the addition of 5 mM dithiothreitol for 30 min @37˚C and then 15 mM iodoacetamide for 30 min @ RT. 4μg trypsin/LysC mix (Promega) were then added and incubated for 4 h @37˚C. 750 μL of Tris-HCl pH 8.0 were then added and incubated for a further 16 h at 37˚C. The digestion was terminated by the addition of 4 μL of TFA. The resultant peptide mixture was purified using HLB prime reverse phase μ-elution plates (Waters) by elution in 70% acetonitrile according to the manufacturers' instructions and dried under vacuum. Peptides were reconstituted in 0.1% formic acid and applied to a Fusion LTQ orbitrap instrument set up as previously described.

**LC-MS/MS analysis of global proteome.** Tryptic peptides were reconstituted in 0.1% formic acid and applied to an Orbitrap Fusion Tribrid Mass Spectrometer with a nano-electrospray ion source as previously described. Peptides were eluted with a linear gradient of 3–8% buffer B (Acetonitrile and 0.1% Formic acid) at a flow rate of 300 nL/min over 5 minutes and then from 8–30% over a further 192 minutes. Full scans were acquired in the Orbitrap analyser using the Top Speed data dependent mode, preforming a MS scan every 3 second cycle, followed by higher energy collision-induced dissociation (HCD) MS/MS scans. MS spectra were acquired at resolution of 120,000 at 300–1,500 m/z, RF lens 60% and an automatic gain control (AGC) ion target value of 4.0e5 for a maximum of 100 ms and an exclusion duration of 40s. MS/MS data were collected in the Ion trap using a fixed collision energy of 32% with a first mass of 110 and AGC ion target of 5.0e3 for a maximum of 100ms.

**Data analysis for proteomics.** Raw data files were analysed using Peaks Studio 10.0 build 20190129. Parent ion tolerance was set to 10ppm and fragment ion tolerance set to 0.6 Da, and spectra were searched against the same database as used for immunopeptidomics. Fixed carbamidomethylation, variable N-terminal acetylation and oxidation of methionine were specified. Variable modifications were set for N-term Acetylation (42.01 Da), Methionine oxidation (15.99 Da) and fixed carboxyamidomethylation modification (57.02 Da) of cysteine. A maximum of three variable modifications per peptide were set. The false discovery rate (FDR) was estimated with decoy-fusion database searches and were filtered to 1% FDR. Relative protein quantification was performed using Peaks software and normalized between samples using a histone ruler [58]. Downstream analysis and visualizations were performed in R using associated packages [26,55–57].

The mass spectrometry proteomics data have been deposited to the ProteomeXchange Consortium via the PRIDE [59] partner repository, see data availability statement.

## Supporting information

**S1 Table. HLA allotypes of the cell lines, lung tissues and dendritic cells used in the study.**
(PDF)

**S2 Table. Summary of immunopeptidomes isolated from cell lines, lung tissues and dendritic cells.** RsPa = resected lung parenchyma tissue, MoDC = monocyte-derived dendritic cells differentiated in vitro from PBMCs from donor P4.
(PDF)

**S3 Table. Immunopeptides isolated from A549 cells following infection with A/Wisconsin/67/2005 influenza.** nM is NetMHC 4.0 predicted binding affinity.
(PDF)

**S4 Table. Influenza peptides identified in Immune Epitope Database (2021-11-03) and from immunopeptidome experiments.**
(PDF)

**S1 Fig. Flow cytometric identification of infected THP1MF 24 h post-infection.** THP1MF were enzymatically dispersed from the culture surface prior to fixation and permeabilization of the cells. Cells were stained intracellularly with FITC-conjugated anti-nucleoprotein antibodies and analysed by flow cytometry. Infected cells were gated with respect to cells exposed to UV-inactivated virus as controls. Figure shown is representative of three independent experiments.
(PDF)

**S2 Fig. Relative proportion of viral proteins in A/H3N2/Wisconsin stock.** Viral proteins were analysed by mass spectrometry proteomics, and the relative quantities of each protein was determined as described in the methods. Quantities are expressed as the percentage of the intensity of the top 3 peptides from each protein from 3 technical replicates.
(PDF)

**S3 Fig. Infection rates in A549 cells prior to DC engulfment.** A549 cells were infected at an MOI of 1.0 for 12 h, achieving >80% infection (Data are mean infection rates from replicate samples stained independently, n = 3 +/- SD).
(PDF)

**S4 Fig. Flow cytometric gating strategy for identification of influenza infection in lung tissue sample cell subsets.** Lung tissue explants were infected with IFV-A ex-vivo and incubated

post-infection for 24 h. Tissue samples were then enzymatically dispersed and the cells stained with monoclonal antibodies conjugated to cell-specific markers. Cell markers were used to identify **(A)** leukocytes (CD45-Horizon). **(B)** CD45+ cells were then gated to identify and exclude T cells (CD3-PECy7). **(C)** CD45+/CD3-/HLA-DR+ cells were macrophages. **(D)** CD45-CD326+ cells were identified as epithelial cells. **(E)** NP1/FITC staining was then used to identify infected macrophages (HLA-DR-APC/Cy7+/FITC+) and epithelial cells **(F)** (CD45-/CD326-PerCP/Cy5.5+/NP-FITC+).
(PDF)

## Acknowledgments

The authors would like to thank the staff at the Flow cytometry Core facility within the Faculty of Medicine at the University of Southampton for their assistance with this work. We would also like to thank Ben Johnson and Katie McCann in the Wessex Investigational Science Hub (WISH) for their assistance with patient samples.

## Author Contributions

**Conceptualization:** Ben Nicholas, Paul Skipp.

**Data curation:** Alistair Bailey.

**Formal analysis:** Alistair Bailey.

**Investigation:** Ben Nicholas.

**Methodology:** Ben Nicholas.

**Project administration:** Ben Nicholas, Tom Wilkinson, Tim Elliott, Paul Skipp.

**Resources:** Ben Nicholas, Karl J. Staples, Tom Wilkinson, Paul Skipp.

**Software:** Alistair Bailey.

**Supervision:** Tom Wilkinson, Tim Elliott, Paul Skipp.

**Visualization:** Ben Nicholas, Alistair Bailey.

**Writing – original draft:** Ben Nicholas, Alistair Bailey.

**Writing – review & editing:** Ben Nicholas, Alistair Bailey, Karl J. Staples, Tom Wilkinson, Tim Elliott, Paul Skipp.

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
