## [Decision Letter · Decision Letter 0]

28 Oct 2021

Dear Dr Bailey,

Thank you very much for submitting your manuscript "Immunopeptidomic analysis of influenza A virus infected human tissues identifies internal proteins as a rich source of HLA ligands" for consideration at PLOS Pathogens. As with all papers reviewed by the journal, your manuscript was reviewed by members of the editorial board and by several independent reviewers. In light of the reviews (below this email), we would like to invite the resubmission of a significantly-revised version that takes into account the reviewers' comments.

Please make certain that your revision addresses the experiments suggested by Reviewer 1. The comparison between the algorithms and your experimental data should be completed. Please expand your discussion of the statistical power of your conclusions based on the number of individuals examined. Finally, if it is feasible to demonstrate that alveolar macrophages are the APC of relevance during influenza please show those data.

We cannot make any decision about publication until we have seen the revised manuscript and your response to the reviewers' comments. Your revised manuscript is also likely to be sent to reviewers for further evaluation.

Sincerely,

Terri M Laufer

Guest Editor

PLOS Pathogens

Carolina Lopez

Section Editor

PLOS Pathogens

Kasturi Haldar

Editor-in-Chief

PLOS Pathogens

orcid.org/0000-0001-5065-158X

Michael Malim

Editor-in-Chief

PLOS Pathogens

orcid.org/0000-0002-7699-2064

Thank you for your submission. Please make certain that your revision addresses the experiments suggested by Reviewer 1. The comparison between the algorithms and your experimental data should be completed. Please expand your discussion of the statistical power of your conclusions based on the number of individuals examined. Finally, if it is feasible to demonstrate that alveolar macrophages are the APC of relevance during influenza please show those data.

Reviewer's Responses to Questions

**Part I - Summary**

Reviewer #1: Nicholas et al. describe the identification of influenza peptides presented by HLA molecules using three human cell-based models. Using these models, they were able to identify peptides with motifs specific for multiple HLA allotypes after infection with both a laboratory-adapted (X31) and clinically relevant (Wisconsin) strain. Importantly, by feeding monocyte derived macrophages apoptotic infected cells, they were able to specifically identify HLA-II associated peptides. Their experiments identified M1 as the most common source protein for both HLA-I and HLA-II peptides after influenza infection. These data confirm other experimental data pointing to internal proteins as major sources of CD8 and CD4 T cell epitopes, which will be important for informing rational design of vaccines that can both better activate T cells and provide better coverage given that internal viral proteins tend to be more conserved across strains. The authors also provide verification of two ex vivo infection models which can further be used for the identification of HLA-I and HLA-II binding peptides. This manuscript is intriguing and provides important information for both vaccine design and for understanding the T cell response to infection. The manuscript is well written, though there some minor edits and clarifications that should be made. A comparison between experimentally identified and predicted peptides would further clarify the benefits of the models used and improve the quality of the paper.

Reviewer #2: The authors showed that there is a limited number of immunopeptides which are served as a target of polymorph MHC type I/II molecules by using three different in vitro systems. Using a MoDC model, they also show that viral membrane bound proteins such as NA, HA and M1 are preferentially presented by HLA-II, and that certain regions of these proteins may be more conducive to processing via the HLA-II pathway.

To design new vaccine candidates against influenza, it is essential to detect the most immunodominant epitopes of the virus. These immunodominant epitopes can cause cross-protection against different influenza strains.

**Part II – Major Issues: Key Experiments Required for Acceptance**

Reviewer #1: The authors indicate the limitations of machine-based algorithms to predict the true efficacy of T cell epitopes and the need for in vitro and ex vivo models such as those they use in this manuscript. However, the authors did not compare the influenza peptide profiles they obtained from their experiments with those predicted by an algorithm for the given cells/HLA allotypes. To reiterate the importance of the experimental methods for determination of HLA ligands, it would be good to show whether all of the identified viral peptides were predicted by the algorithm, whether the top predictions from the algorithm were identified in vitro or not, if a higher frequency of predicted peptides were identified using one model or over the other, and if any similarities/differences could be identified between the peptides predicted by the algorithm and those which where or were not identified in vitro. As the authors did note in the manuscript, there are limitations in peptide identification by MS so this has to be taken in consideration when making inferences from the comparison between experimental and theoretical peptide identification, but the comparison should be made.

Reviewer #2: The authors mention alveolar macrophages in the abstract however, there is no direct experiment showing that dendritic cell engulf the apoptotic lung macrophages. Moreover, the role and the presence of interstitial macrophages should be also considered and dissected. Using the available FACS Aris instrument, the infected populations of antigen presenting cells should be determined.

The classical HLA alleles are the most polymorphic in the human population. The authors should involve more donors to support the conclusions of the manuscript.

**Part III – Minor Issues: Editorial and Data Presentation Modifications**

Reviewer #1: - In lines 172-176, the authors describe the number of peptides and proteins observed for each participant without context that these are host derived, as was done for THP1M� infection in lines 217-219. As the authors describe the virus-specific peptides in the lines immediately preceding the proteomic data, the natural assumption of the reader is that they are continuing to discuss virus-specific findings if clarification is not given.

- Lines 185-187 would benefit from some clarification of the methods to identify allotypes and motifs. If I have understood correctly, you used unbiased cluster analysis of the observed host peptides from each sample to derive the 9-mer binding motifs for each HLA allotype (shown in Fig 2 C-E), and then used these binding motifs to match influenza peptides with HLA allotype (as indicated in Table 1). This is not directly clear from the description. To this effect, the figure legend for Fig. 2 C-E should just say “Class I HLA allotype 9-mer binding motifs derived from host immunopeptides using unsupervised clustering using MixMHCp.”

- In lines 248-249, the authors suggest that the majority of viral peptides were predicted to be strong binders to the HLA-B, however the majority of the peptides in Table 2 (including all HLA-B associated peptides) do not have predicted affinities to indicate the strength of their binding. Can the authors clarify how they are defining strong binders?

- In Table 2, there are no peptides in blue to indicate mis-matched aa sequences.

- In Table 2, at first glance it looks like there are 2 unique peptides after Wisconsin infection, but this is because the NS1 peptide is listed twice to allow for reporting of the different predicted binding for the two virus strains. While this is understable, it would be helpful if those two peptides were at the very least listed directly after one another if not somehow noted to be the same.

- It is noted that nM is predicted affinity in Table S2, but this is not indicated in any of the main tables in the manuscript.

- Table 3 has two “motif” columns.

- Table S2 and Table S3 are switched, Table S2 should be the the summary of immunopeptidomes from cell lines, lung tissue, and DCs whereas Table S3 should be the immunopeptidome of A549 cells

- There are some instances in which the wrong Table is referred to: Line 237 should be Table 2 and Line 292 should be Table S2

- There is an extra comma at the end of line 86 in the phrase “acquired infections, which also”

- In the methods for the phagocytosis assay, the authors mention monocyte generation media and DC activation supplement (line 566-567) but do not identify the components of these solutions or the source if purchased directly from a company.

Reviewer #2: The purity of moDC cultures should be measured by the expression levels of CD1a(+), CD209(+), MHCII(high) and CD14(-).

Figure S1: The authors show one representative of three independent experiments. Please, incorporate a diagram showing the mean values +/- SD of the experiments.

Some minor mistyping issues are detected, please revise the manuscript to exclude such minor errors.

PLOS authors have the option to publish the peer review history of their article (what does this mean?). If published, this will include your full peer review and any attached files.

Reviewer #1: No

Reviewer #2: No
---

## [Editor Report · Decision Letter 1]

2 Jan 2022

Dear Dr Bailey,

We are pleased to inform you that your manuscript 'Immunopeptidomic analysis of influenza A virus infected human tissues identifies internal proteins as a rich source of HLA ligands' has been provisionally accepted for publication in PLOS Pathogens.

Best regards,

Terri M Laufer

Guest Editor

PLOS Pathogens

Carolina Lopez

Section Editor

PLOS Pathogens

Kasturi Haldar

Editor-in-Chief

PLOS Pathogens

orcid.org/0000-0001-5065-158X

Michael Malim

Editor-in-Chief

PLOS Pathogens

orcid.org/0000-0002-7699-2064
---

## [Editor Report · Acceptance letter]

14 Jan 2022

Dear Dr Bailey,

We are delighted to inform you that your manuscript, "Immunopeptidomic analysis of influenza A virus infected human tissues identifies internal proteins as a rich source of HLA ligands," has been formally accepted for publication in PLOS Pathogens.

Best regards,

Kasturi Haldar

Editor-in-Chief

PLOS Pathogens

orcid.org/0000-0001-5065-158X

Michael Malim

Editor-in-Chief

PLOS Pathogens

orcid.org/0000-0002-7699-2064